# FunkNN: Neural Interpolation for Functional Generation

**AmirEhsan Khorashadizadeh**[*], **Anadi Chaman**[†], **Valentin Debarnot**[*] **and Ivan Dokmanić**[*†]

[*]Department of Mathematics and Computer Science, University of Basel
[†]Coordinated Science Laboratory, University of Illinois at Urbana-Champaign

## Abstract

Can we build continuous generative models which generalize across scales, can be evaluated at any coordinate, admit calculation of exact derivatives, and are conceptually simple? Existing MLP-based architectures generate worse samples than the grid-based generators with favorable convolutional inductive biases. Models that focus on generating images at different scales do better, but employ complex architectures not designed for continuous evaluation of images and derivatives. We take a signal-processing perspective and treat continuous image generation as interpolation from samples. Indeed, correctly sampled discrete images contain all information about the low spatial frequencies. The question is then how to extrapolate the spectrum in a data-driven way while meeting the above design criteria. Our answer is FunkNN—a new convolutional network which learns how to reconstruct continuous images at arbitrary coordinates and can be applied to any image dataset. Combined with a discrete generative model it becomes a functional generator which can act as a prior in continuous ill-posed inverse problems. We show that FunkNN generates high-quality continuous images and exhibits strong out-of-distribution performance thanks to its patch-based design. We further showcase its performance in several stylized inverse problems with exact spatial derivatives. Our implementation is available at https://github.com/swing-research/FunkNN.

## 1 Introduction

Deep generative models are effective image priors in applications from ill-posed inverse problems (Shah & Hegde, 2018; Bora et al., 2017) to uncertainty quantification (Khorashadizadeh et al., 2022) and variational inference (Rezende & Mohamed, 2015). Since they approximate distributions of images sampled on discrete grids they can only produce images at the resolution seen during training. But natural, medical, and scientific images are inherently continuous. Generating continuous images would enable a single trained model to drive downstream applications that operate at arbitrary resolutions. If this model could also produce exact spatial derivatives, it would open the door to generative regularization of many challenging inverse problems for partial differential equations (PDEs).

There has recently been considerable interest in learning grid-free image representations. Implicit neural representations (Tancik et al., 2020; Sitzmann et al., 2020; Martel et al., 2021; Saragadam et al., 2022) have been used for mesh-free image representations in various inverse problems (Chen et al., 2021; Park et al., 2019; Mescheder et al., 2019; Chen & Zhang, 2019; Vlašić et al., 2022; Sitzmann et al., 2020). An implicit network $f_\theta(\mathbf{x})$, often a multi-layered perceptron (MLP), directly approximates the image intensity at spatial coordinate $\mathbf{x} \in \mathbb{R}^D$. While $f_\theta(\mathbf{x})$ only represents a single image, different works incorporate a latent code $\mathbf{z}$ in $f_\theta(\mathbf{x}, \mathbf{z})$ to model *distributions* of continuous images.

These approaches perform well on simple datasets but their performance on complex data like human faces is far inferior to that of conventional grid-based generative models based on convolutional neural networks (CNNs) (Chen & Zhang, 2019; Dupont et al., 2021; Park et al., 2019). This is in fact true even when evaluated at resolution they were trained on. One reason for their limited performance is that these implicit models use MLPs which are not well-suited for modelling image data.

Figure 1: The proposed architecture. The generative model (orange) produces a fixed-resolution image that is differentiably used by FunkNN to produce the image intensity at any location (blue).

In addition, unlike their grid-based counterparts, these implicit generative models suffer from a significant overhead in the form of a separate encoding hyper-network (Chen & Zhang, 2019; Chen et al., 2021; Dupont et al., 2021) or parameter training (Park et al., 2019) to obtain latent codes $\mathbf{z}$.

In this paper, we alleviate the above challenges with a new mesh-free convolutional image generator that can faithfully learn the distribution of continuous image functions. The key feature of the proposed framework is our novel patch-based continuous super-resolution network—FunkNN— which takes a discrete image at any resolution and super-resolves it to generate image intensities at arbitrary spatial coordinates. As shown in Figure 1), our approach combines a traditional discrete image generator with FunkNN, resulting in a deep generative model that can produce images at arbitrary coordinates or resolution. FunkNN can be combined with *any* off-the-shelf pre-trained image generator (or trained jointly with one). This includes the highly successful GAN architectures (Karras et al., 2019; 2020), normalizing flows (Kingma & Dhariwal, 2018; Kothari et al., 2021) or score-matching generative models (Song & Ermon, 2019; Song et al., 2020). It naturally enables us to learn complex image distributions.

Unlike prior works (Chen & Zhang, 2019; Chen et al., 2021; Dupont et al., 2021; Park et al., 2019), FunkNN neither requires a large encoder to generate latent codes nor does it use any MLPs. This is possible thanks to FunkNN's unique way of integrating the coordinate $\mathbf{x}$ with image features. The key idea is that resolving image intensity at a coordinate $\mathbf{x}$ should only depend on its neighborhood. Therefore, instead of generating a code for the entire image and then combining it with $\mathbf{x}$ using an MLP, FunkNN simply crops a patch around $\mathbf{x}$ in the low-resolution image obtained from a traditional generator. This window is then provided to a small convolutional neural network that generates the image intensity at $\mathbf{x}$. The window cropping is performed in a differentiable manner with a spatial transformer network (Jaderberg et al., 2015).

We experimentally show that FunkNN reliably learns to resolve images to resolutions much higher than those seen during training. In fact, it performs comparably to state-of-the-art continuous super-resolution networks (Chen et al., 2021) despite having only a fraction of the latter's trainable parameters. Unlike traditional learning-based methods, our approach can also super-resolve images that belong to image distributions different from those seen while training. This is a benefit of patch-based processing which reduces the chance of overfitting on global image features. In addition, we show that our overall generative model framework can produce high quality image samples at any resolution. With the continuous, differentiable map between spatial coordinates and image intensities, we can access spatial image derivatives at arbitrary coordinates and use them to solve inverse problems.

## 2    IMPLICIT NEURAL REPRESENTATIONS FOR CONTINUOUS IMAGE REPRESENTATION

Let $u \in L^2(\mathbb{R}^D)^c$ denote a continuous signal of interest with $c \geq 1$ channels and $\mathbf{u}$ be its discretized version supported along a fixed grid with $n \times n \in \mathbb{N}^*$ elements. For simplicity we consider the case with $D = 2$. Our discussion naturally extends to higher dimensions.

An implicit neural representation approximates $u$ by $f_{\boldsymbol{\theta}} : \mathbb{R}^2 \to \mathbb{R}$ parameterized by weights $\boldsymbol{\theta} \in \mathbb{R}^K$. In standard approaches, the weights are obtained by solving

$$\boldsymbol{\theta}^*(u) = \arg\min_{\boldsymbol{\theta}} \sum_{i=1}^{n^2} \|f_{\boldsymbol{\theta}}(\mathbf{x}_i) - \mathbf{u}(\mathbf{x}_i)\|_2^2, \tag{1}$$

where $\mathbf{u}(\mathbf{x}_i)$ is the image target value sampled at spatial coordinate $\mathbf{x}_i$. This way, implicit neural representations can provide a continuous interpolation for $\mathbf{u}$. Notice, however, that the properties of this interpolation scheme depend on the choice of the network architecture and do not exploit the statistics of given images.

From (1), we can observe that the obtained weights $\boldsymbol{\theta}^*(u)$ apply only to a single image $\mathbf{u}$, thereby making $f_\theta(\mathbf{x})$ incapable of providing interpolating representations for a class of images. Some prior works address this by incorporating a low dimensional latent code $\mathbf{z}$ in the implicit network. The new representation then is given by the map $\mathbf{x} \mapsto f_{\boldsymbol{\theta}}(\mathbf{x}, \mathbf{z})$, where each $\mathbf{z}$ represents a different image. Note that this framework often entails an additional overhead to relate the images with the latent codes. For example, (Chen & Zhang, 2019) and (Dupont et al., 2021) jointly train $f_\theta$ with an encoder network to generate the latent codes for images. (Park et al., 2019), on the other hand, optimize the codes as trainable parameters. Figure 2 shows a general illustration of implicit generative networks.

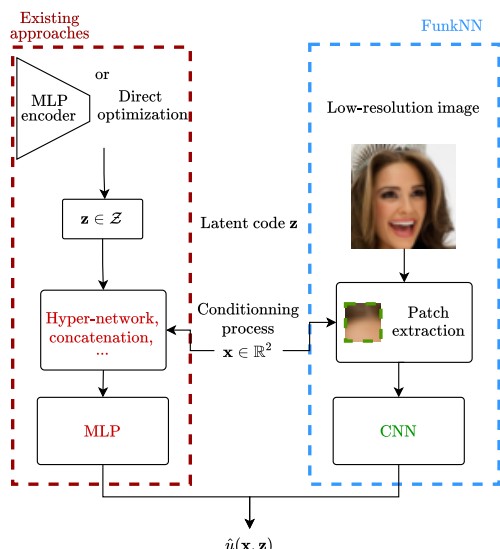

Figure 2: Conceptual difference between FunkNN and existing implicit neural representations.

The prior approaches to model continuous image distributions often exhibit poor performance in comparison to their grid-based generative counterparts (Chen & Zhang, 2019; Dupont et al., 2021). This is due to their reliance on MLP networks which a priori do not possess the right inductive biases to model images.

## 3 OUR APPROACH

To address the challenges discussed in Section 2 we propose a convolutional image generator that produces continuous images. As shown in Figure 1, the proposed framework relies on FunkNN, our new continuous super-resolution network which interpolated images to arbitrary resolution. Our approach first samples discrete fixed-resolution images from an expressive *discrete* convolutional generator. This is followed by continuous interpolation with FunkNN. We combine FunkNN with any state-of-the-art pre-trained image to learn complex image distributions.

The key advantage of FunkNN is its architectural simplicity and interpretability from a signal processing perspective. The main idea is that super-resolving a sharp feature from a low-resolution image at a spatial coordinate $\mathbf{x}$ should only use information from the neighborhood of $\mathbf{x}$. This is because high frequency structures like edges are spatially localized and have small correlations with pixels at far-off spatial locations in the low-resolution image. FunkNN, therefore, selects a small patch around coordinate $\mathbf{x}$ and passes it to a small CNN to regress image intensity at $\mathbf{x}$. Figure 2 illustrates the differences between FunkNN and prior approaches. In the following sections, we provide a formal description of FunkNN and our continuous image generative framework.

### 3.1 FUNKNN: CONTINUOUS SUPER-RESOLUTION NETWORK

Given a low-resolution image $\mathbf{u}_{lr} \in \mathbb{R}^{d \times d \times c}$, FunkNN estimates the image intensity at a spatial coordinate $\mathbf{x} \in \mathbb{R}^2$ by a two-step process: i) it first extracts a $p \times p$ patch $\mathbf{w} \in \mathbb{R}^{p \times p \times c}$ from $\mathbf{u}$ around

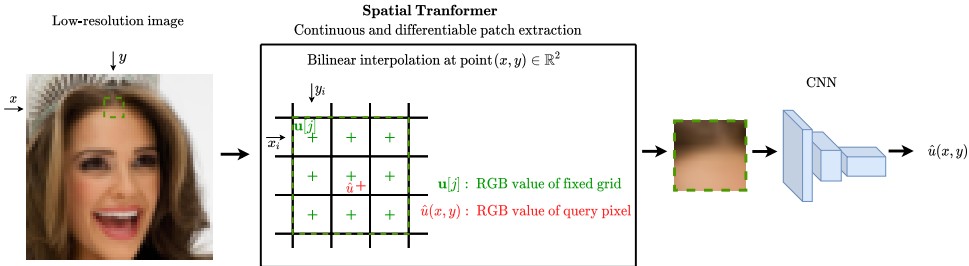

Figure 3: FunkNN architecture. Given a location $(x, y)$, the Spatial Transformer extracts a patch in the discrete image and a CNN produces the intensity at the query position.

the coordinate $\mathbf{x}$, and then ii) a CNN maps the patch to the estimated intensity. More formally, if we denote $\mathrm{ST} : (\mathbf{x}, \mathbf{u}) \mapsto \mathbf{w}$ to be a patch extraction module and $\mathrm{CNN}_{\boldsymbol{\theta}} : \mathbb{R}^{p \times p \times c} \to \mathbb{R}^c$ to be a CNN block that estimates pixel intensity at $\mathbf{x}$ from $\mathbf{w}$, then FunkNN can be characterized as

$$\mathrm{FunkNN}_{\boldsymbol{\theta}} \overset{\text{def.}}{=} \mathrm{CNN}_{\boldsymbol{\theta}} \circ \mathrm{ST}.$$

While we could extract a patch in the ST module by trivially cropping it from the image, doing so would not give access to derivatives of image intensities with respect to spatial coordinates, a property of importance for PDE-based problems. We address this by modelling ST with a spatial transformer network (Jaderberg et al., 2015). As illustrated in Figure 3, spatial transformers map the low-resolution image and input coordinate to the cropped patch in a continuous manner, thereby preserving differentiability. These derivatives can, in fact, be exactly computed using auto-differentiation in PyTorch (Paszke et al., 2019) and TensorFlow (Abadi et al., 2016). We evaluate the accuracy of these derivatives in Figure 13 in the Appendix E. This experiment clearly shows the effectivity of FunkNN for approximating the first-order and second-order derivatives, crucially important for solving PDE-based inverse problems. Another advantage of the ST module over trivial cropping is that it allows us to learn the effective receptive field of the patches during training, thereby providing better performance. For further details, please refer to the Appendix A.2.

### 3.1.1 TRAINING FUNKNN

Let $(\mathbf{u}_m)_{1 \leq m \leq M}$ denote a dataset of $M$ discrete images with the maximum resolution $n \times n$. We generate low-resolution images $(\mathbf{u}_{\mathrm{lr},m})_{1 \leq m \leq M}$ from this dataset where $\mathbf{u}_{\mathrm{lr},m} \in \mathbb{R}^{d \times d \times c}$ and $d < n$. FunkNN is then trained to super-resolve $(\mathbf{u}_{\mathrm{lr},m})$ with $(\mathbf{u}_m)$ as targets using a patch-based framework. In particular, from each low-resolution image, small $p \times p$ sized patches centered at randomly selected coordinates are extracted and provided to FunkNN which then regresses the corresponding intensity at the coordinates in the high-resolution image. We optimize the weights of the CNN as follows.

$$\underset{\boldsymbol{\theta} \in \mathbb{R}^K}{\arg\min} \sum_{m=1}^{M} \sum_{i=1}^{n^2} \left| \mathrm{FunkNN}_{\boldsymbol{\theta}}(\mathbf{x}_i, \mathbf{u}_{\mathrm{lr},m}) - u_m(\mathbf{x}_i) \right|^2. \tag{2}$$

Note that since FunkNN takes patches as inputs, it can be used to simultaneously train on images with different sizes. In addition, it allows us to train with mini-batches on both images and sampling locations, thereby enabling memory-efficient training.

We consider three strategies to train our model, namely (i) *single*, (ii) *continuous* and (iii) *factor*. In *single* mode, FunkNN is trained with input images of fixed resolution, eg. $d = \frac{n}{2}$. The *continuous* mode uses low-resolution input images at different scales, i.e. $d = \frac{n}{s}$, where the scale $s$ can vary between $[s_{\min}, s_{\max}]$. In *factor* mode, the super-resolution factor, $s$, between the low and high-resolution images is always fixed and the low-resolution size $d$ is varied in the range $d_{\min}$ and $d_{\max}$. The high-resolution target size is then correspondingly chosen as $ds \times ds$. At test time, we can then use this model to super-resolve images by a factor $s$ in a hierarchical manner. For instance, with

input image $\mathbf{u}_{hr}^{(0)} = \mathbf{u}$ of size $d \times d$, it can iteratively generate high-resolution images $(\mathbf{u}_{hr}^{(i)})_{i \geq 1}$ of sizes $(d_i)_{i \geq 1}$ satisfying $\mathbf{u}_{hr}^{(i)} = \text{FunkNN}_\theta(\mathbf{u}_{hr}^{(i-1)})$ and $d_i = s^i d$.

## 3.2 GENERATIVE NETWORK

To generate continuous samples, we can combine FunkNN with *any* fixed-resolution convolutional generator. This flexibility allows our overall model to learn complex continuous image distributions, and to produce images and their spatial derivatives at arbitrary resolutions. More formally, let $G : \mathbb{R}^L \to \mathbb{R}^{d \times d \times c}$ be a generative prior on low-resolution images of size $d \times d$ that takes as input a latent code of size $L$. Then, $G$ can be combined with $\text{FunkNN}_\theta$ to yield a continuous image generator $\text{FunkNN}_\theta \circ G$.

The choice of fixed-sized generator we use prior to FunkNN depends on the downstream application. Generative adversarial networks (GANs) (Karras et al., 2019) and normalizing flows (Kingma & Dhariwal, 2018) are popular high-quality image generators. However, the former exhibits challenges when used as generative prior for inverse problems (Bora et al., 2017; Kothari et al., 2021) and the latter have very large memory requirements and are expensive to train (Whang et al., 2021; Asim et al., 2020; Kothari et al., 2021). Injective encoders (Kothari et al., 2021) were recently proposed to alleviate these challenges by using a small network and latent space. However, since we do not necessarily require injectivity in our applications of focus, we use a simpler architecture inspired from (Kothari et al., 2021) for faster training. In particular, we use a conventional auto-encoder to obtain a latent space for our training images and use a low-dimensional normalizing flow to learn this latent space distribution. The standard practice of using mean squared loss as suggested in (Brehmer & Cranmer, 2020; Kothari et al., 2021) results in a loss of high frequency components in our reconstructions. We remedy this by using a perceptual loss (Johnson et al., 2016). Further details on network architecture and training are given in Appendix A.1. It is worth noting that if generative modelling is not aimed, we can train FunkNN independently of any generator as shown in Section 5.1.

## 4 CONTINUOUS GENERATIVE MODELS AND SOLVING INVERSE PROBLEMS

Continuous generative models built using FunkNN can be used for image reconstruction at resolutions much higher than those seen during training. This is particularly useful when solving ill-posed inverse problems commonly found in experimental sciences like seismic (Bergen et al., 2019), medical (Lee et al., 2017) and 3-D molecular imaging (Arabi & Zaidi, 2020). Due to difficulty and high cost of measurement acquisition, obtaining training data is challenging in these applications. Therefore, new models can not readily be trained when faced with measurements acquired at resolution different from the ones previously seen during training.

We consider three inverse problems: reconstructing an image given its (i) first derivatives, (ii) when a fraction of derivatives are known, and (iii) limited-view computed tomography (CT). In the following, we assume that we have low-resolution generative prior on $d \times d$-pixel images, $G : \mathbb{R}^L \to \mathbb{R}^{d \times d \times c}$, which takes as input a latent code $\mathbf{z} \sim \mathcal{N}(0, \text{Id}_L)$.

### 4.1 INVERTING DERIVATIVES

We first apply FunkNN to reconstruct continuous image $u$ from its spatial derivatives $\nabla_{\mathbf{x}} u(\mathbf{x}_j)$ in coordinates $\{\mathbf{x}_j\}_{j=1}^{n^2}$. Related ill-posed reconstructions appear in different domains such as computer vision (Park et al., 2019) or obstacle scattering (Vlašić et al., 2022). We use the exact spatial derivatives provided by FunkNN and a pre-trained generative model to obtain the latent code $\mathbf{z}^*$ that gives us output aligned with the given derivatives,

$$\mathbf{z}^* = \arg\min_{\mathbf{z}} \sum_{j=1}^{n^2} \|\nabla_{\mathbf{x}} \text{FunkNN}_{\boldsymbol{\theta}}(\mathbf{x}_j, G(\mathbf{z})) - \nabla_{\mathbf{x}} u(\mathbf{x}_j)\|_2^2 + \lambda \|\mathbf{z}\|_2^2. \tag{3}$$

We also found that updating the weights $\boldsymbol{\theta}$ of the generative model further helps in producing better reconstruction as suggested in (Hussein et al., 2020). The exact procedure is detailed in Appendix B.1. The estimated reconstruction is given by sampling the trained FunkNN network on the high-resolution grid: $\hat{u}(\mathbf{x}_j) = \text{FunkNN}_{\boldsymbol{\theta}}(\mathbf{x}_j, G(\mathbf{z}^*))$.

## 4.2 SPARSE DERIVATIVES

We now make the previous problem even more ill-posed by observing not the spatial derivatives on a dense grid, but only 20% of them that have the highest intensity. It amounts to only sample the gradient on the $n_s < n^2$ locations that gives the largest value $\|\nabla_\mathbf{x} u(\mathbf{x}_j)\|$. In order to account for the missing low-amplitude gradient, we add an extra total-variation regularization term. Thus, we aim at solving

$$\mathbf{z}^* = \arg\min_\mathbf{z} \sum_{j=1}^{n_s} \|\nabla_\mathbf{x}\text{FunkNN}_{\boldsymbol{\theta}}(\mathbf{x}_j, G(\mathbf{z})) - \nabla_\mathbf{x} u(\mathbf{x}_j)\|_2^2 + \lambda\|\mathbf{z}\|_2^2 + \lambda_2\|\nabla G(\mathbf{z})\|_2, \quad (4)$$

using the same process as in the previous setting (see Appendix B.1).

## 4.3 LIMITED-VIEW CT

In CT, we observe projections of a 2-dimensional image $u \in L^2(\mathbb{R}^2)$ through the (discrete angle) Radon operator $\mathcal{A}(\boldsymbol{\alpha}) : L^2(\mathbb{R}^2) \to L^2(\mathbb{R})^I$ where $\boldsymbol{\alpha} = \{\alpha_1, \ldots, \alpha_I\}$ denotes the $I$ viewing directions with $\alpha_i \in [0, 2\pi)$. Finally, the discrete observation is given by

$$\mathbf{v}_i = (\mathcal{A}(\alpha_i)u)(\mathbf{x}) + \boldsymbol{\eta}_i, \quad (5)$$

where $\mathbf{x} = [\mathbf{x}_1, \ldots, \mathbf{x}_{n^2}]$ is the $n \times n$ sampling grid and $\boldsymbol{\eta}_i \overset{\text{iid}}{\sim} \mathcal{N}(0, \text{Id}_{n^2})$ is a random perturbation. If the view directions correspond to a fine sampling of the interval $[0, 2\pi)$, then the adjoint operator $\mathcal{A}^*$, namely the filtered back-projection (FBP) operator, allows to recover the original signal even in the presence of noise. For application-specific reasons it is not always possible to observe the full viewing direction set, leading to limited-view CT. Here we sample $(\alpha_i)$ uniformly between $-70$ degrees and $+70$ degrees. This incomplete set of observations makes the estimation of the signal $u$ from (5) strongly ill-posed. We address limited-view CT using the FunkNN framework:

$$\mathbf{z}^* = \arg\min_\mathbf{z} \sum_{i=1}^{I} \|\mathcal{A}(\alpha_i)\text{FunkNN}_{\boldsymbol{\theta}}(\mathbf{x}, G(\mathbf{z})) - \mathbf{v}_i\|_2^2 + \lambda\|\mathbf{z}\|_2^2, \quad (6)$$

## 5 EXPERIMENTAL RESULTS

### 5.1 SUPER-RESOLUTION

We first evaluate the continuous, arbitrary-scale super-resolution performance of FunkNN. We work on the CelebA-HQ (Karras et al., 2017) dataset with resolution $128{\times}128$, and we train the model using three training strategies explained in section 3.1.1, *single*, *continuous* and *factor*. More details about network architecture and training details are provided in Appendix A.2.

The model *never sees training data at resolutions 256 × 256 or higher*. Figure 4 shows the performance of FunkNN trained using *factor* strategy, which indicates that FunkNN yields high-quality image reconstructions at much higher resolutions than it was trained on. This experiment shows that FunkNN learns an accurate super-resolution operator and generalizes to resolutions not seen during training. Table 1 shows that our model demonstrates comparable performance to the state-of-the-art baseline LIIF-EDSR (Chen et al., 2021) but with a considerably smaller network; our model only has 140K trainable parameters while LIIF uses 1.5M parameters. Additionally, we provide a comparison with LIIF by pruning its trainable parameters to reach 140k. The results suggest that while FunkNN obtain comparable results to the original LIIF architecture, reducing its parameters to be closer to FunkNN deteriorates the results. This experiment emphasizes the capacity of FunkNN to perform similarly to state-of-the-art method wile being conceptually simpler. It is worth mentioning that on the same Nvidia A100 GPU, the average time to process 100 images of size $128 \times 128$ is 1.97 seconds for LIIF, 0.81 seconds for LIIF with 140k parameters and 0.7 seconds for FunkNN.

### 5.2 ROBUSTNESS TO OUT-OF-DISTRIBUTION SAMPLES

One exciting aspect of the proposed model is its strong generalization to out-of-distribution (OOD) data–a boon obtained from FunkNN's patch-based operation. We showcase it by evaluating FunkNN

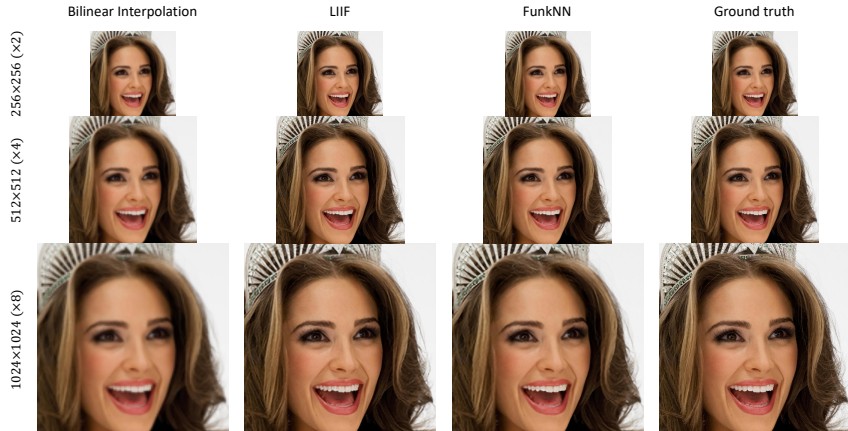

Figure 4: FunkNN model is only trained over CelebA-HQ images of maximum resolution $128 \times 128$, but it can reliably reconstruct high-quality images in higher resolutions.

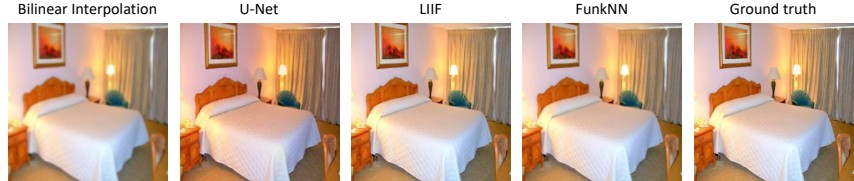

Figure 5: Image super-resolution on $256 \times 256$ LSUN images with the model trained on $128 \times 128$ images from CelebA-HQ data.

originally trained on CelebA-HQ (Karras et al., 2017) at resolution $128 \times 128$ (Section 5.1), to super-resolve images from the LSUN-bedroom (Yu et al., 2015) dataset which are structurally very different from the training data. Moreover, we super-resolve images from size 128 to 256, which were not seen at training time. Figure 5 indicates that the proposed model can faithfully super-resolve OOD images at new resolutions. We compare our approach with U-Net (Ronneberger et al., 2015) which has shown promising results for super-resolution; however, it can only be trained to map between two fixed resolutions. We trained a U-Net to superresolve from $128 \times 128$ to $256 \times 256$; it was thus trained on larger images than other methods. Nevertheless, Table 1 shows that while the U-Net performs well on inlier samples, it gives rather bad results for OOD data. We also report the shift-invariant SNRs (Weiss et al., 2019) in Table 2 to compensate for the fact that U-Net is unable to recover the correct amplitudes for OOD samples.

Table 1: Performance of different methods over super-resolution in SNR (dB) ; the SNRs are averaged over 100 test samples of CelebA-HQ (inliers) or LSUN-bedroom (outliers).

| | $128 \rightarrow 256$ | $128 \rightarrow 512$ | $128 \rightarrow 1024$ | $128 \rightarrow 256$ (OOD) | Trainable params |
|---|---|---|---|---|---|
| *Bilinear Interpolation* | 27.5 | 24.9 | 22.4 | 22.9 | - |
| *U-Net (Ronneberger et al., 2015)* | 31.3 | - | - | 19.9 | 8800K |
| *LIIF-EDSR (Chen et al., 2021)* | 31.6 | 28.0 | 23.9 | **28.0** | 1500K |
| *LIIF-EDSR (Chen et al., 2021) (140K param.)* | 31.6 | 27.5 | 23.8 | 26.7 | 140K |
| *FunkNN (single)* | 31.2 | 26.3 | 22.5 | 27.2 | 140K |
| *FunkNN (continuous)* | 30.7 | 27.7 | 23.5 | 26.4 | 140K |
| *FunkNN (factor)* | **32.6** | **28.2** | **24.2** | 26.6 | 140K |

## 5.3 INVERSE PROBLEMS

We solve equation 3 using the generative framework described in Section 4 trained over CelebA-HQ samples to reconstruct an image from its spatial derivatives. Experimental settings are provided in Appendix B.1. Figure 6 illustrates FunkNN's performance, which provides high-quality spatial

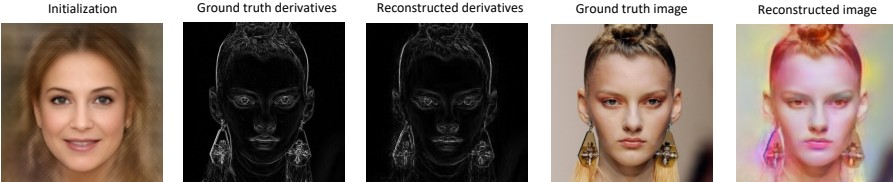

Figure 6: Reconstructing an image from its spatial derivatives using the generative prior; FunkNN could faithfully reconstruct the image details.

reconstructions from the observation of the spatial derivatives. Notice that Problem (3) is ill-posed and it is not possible to accurately estimate mean intensity of the original image only based on the derivatives. Despite the different color scales, the unknown image is accurately estimated by FunkNN.

We solve equation 4 with the same procedure but over LoDoPaB-CT samples (Leuschner et al., 2021). We compare FunkNN with the continuous GAN proposed by (Dupont et al., 2021). We solve the problem described in Equation (4) but replacing our model by their GAN. The training procedure is described in Appendix A.3. Figure 7 shows that FunkNN significantly outperforms the continuous GAN; while we observe an accurate estimation of the image gradients by GAN (first row, second column), its reconstruction (second row, second column) lacks a lot of details that were well retrieved by our method. The poor reconstruction of GANs as a prior for solving inverse problems has already been observed in (Whang et al., 2021; Kothari et al., 2021) and is confirmed by this experiment. Shift-invariant SNRs are reported in the bottom-left corner of each reconstruction. This example of highly under-determined system shows that the proposed FunkNN architecture can leverage existing prior in addition to super-resolving both an image and its derivative.

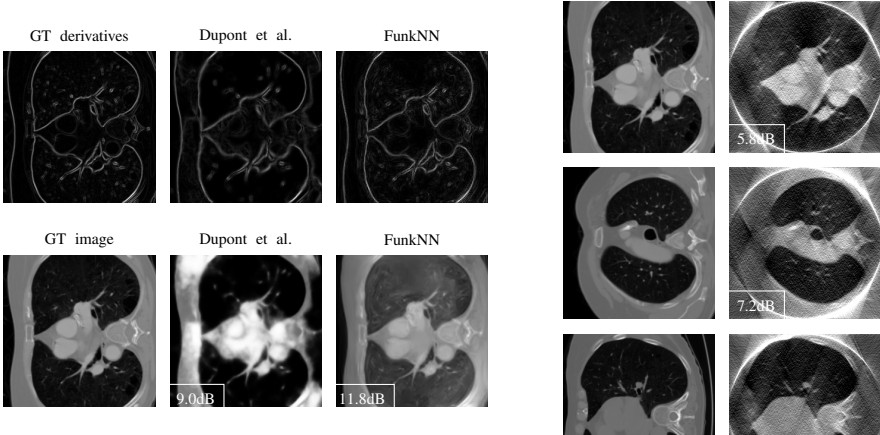

Figure 7: Solving PDE-based inverse problem with sparse derivatives.

Figure 8: Limited-view CT reconstruction.

We repeat the same procedure and solve equation 6 to reconstruct an image from its limited-view sinogram. The precise experimental setting is relegated to Appendix C. Figure 8 shows FunkNN's reconstructions compared to FBPs which confirms the power of the learned low-resolution prior in solving Problem (5).

## 6 RELATED WORK

### 6.1 SUPER-RESOLUTION

Various learning based approaches have been proposed over the years for image super-resolution. Patch-based dictionary learning (Yang et al., 2010) and random forests (Schulter et al., 2015) have been used to learn a mapping between fixed low and high-resolution image pairs. More recently,

deep learning methods based on convolutional neural networks have provided state-of-the-art performance on super-resolution tasks (Dong et al., 2014; 2015; 2016; Kim et al., 2016; Lim et al., 2017; Lai et al., 2017). Inspired by the popular U-Net (Ronneberger et al., 2015), (Liu et al., 2022) proposed a multi-scale convolutional sparse coding based approach that performed comparably to U-Net on various tasks including super-resolution. Furthermore, (Jia et al., 2017) learned the interpolation kernel to design an implicit mapping from low to high resolution samples.

(Bora et al., 2017) and (Kothari et al., 2021) have used generative models as priors for image super-resolution using an iterative process. In addition, conditional generative models have been used to address ill-posedness of super-resolution (Ledig et al., 2017; Wang et al., 2018; Lugmayr et al., 2020; Khorashadizadeh et al., 2022). While the prior works mentioned above exhibit excellent super-resolution performance, they all have a limitation—these models are restricted to be used only at the image resolution seen during training. On the contrary, we focus on super-resolving images to arbitrary resolution.

In an attempt towards building continuous super-resolution models, (Hu et al., 2019) propose a new upsampling module, MetaSR, that adjusts its upscale weights dynamically to generate high resolution images. While this approach exhibits promising results on in-distribution training scales, it has limited performance of out of distribution factors. The closest to our work is the network LIIF (Chen et al., 2021) that also uses local neighbourhoods around spatial coordinates for super-resolution. However, unlike FunkNN that directly operates on small image patches, LIIF first uses a large convolutional encoder to generate feature maps for the image. Image intensity at a coordinate $\mathbf{x}$ is then obtained by passing this coordinate along with features in its neighbourhood to an MLP. Note that FunkNN's approach of combining image features with spatial coordinates is architecturally much simpler than that of LIIF. It also requires less parameters, is slightly faster to evaluate and performs equivalently well.

## 6.2 GENERATIVE MODELS FOR CONTINUOUS SIGNALS

Generative models based on MLPs have been proposed in recent years to learn continuous image distributions. (Chen & Zhang, 2019) use implicit representations to learn 3D shape distributions but lack expressivity to represent complex image data. (Dupont et al., 2021) propose adverserial training based on MLPs and Fourier features (Tancik et al., 2020) to represent different data modalities like faces, climate data and 3D shapes. While their approach is quite versatile, it still performs poorly compared to convolutional image generators (Radford et al., 2015; Karras et al., 2019; 2020).

As a step towards building expressive continuous image generators, (Xu et al., 2021) and (Ntavelis et al., 2022) used positional encodings to make style-GAN generator (Karras et al., 2019) shift and scale invariant, thereby resulting in arbitrary scale image generation. (Skorokhodov et al., 2021; Anokhin et al., 2021) proposed continuous GANs for effectively learning representations of real-world images. However, these models do not provide access to derivatives with respect to spatial coordinates, making them unsuitable for solving inverse problems.

In a related line of work, (Ho et al., 2022) propose to combine a cascade of a diffusion and fixed-size super-resolution models to generate images at higher resolution. Unlike FunkNN, they use fixed-size super-resolution blocks so their approach does not generate continuous images.

## 7 CONCLUSION

Recent continuous generative models rely on fully connected layers and lack the right inductive biases to effectively model images. In this paper, we address continuous generation in two steps. (i) Instead of generating continuous images directly, we first use a convolutional generator pre-trained to sample discrete images from the desired distribution. (ii) We then use FunkNN to continuously super-resolve the discrete images given by the generative model. Our experiments demonstrate performance comparable to state-of-the-art networks for continuous super-resolution despite using only a fraction of trainable parameters.

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

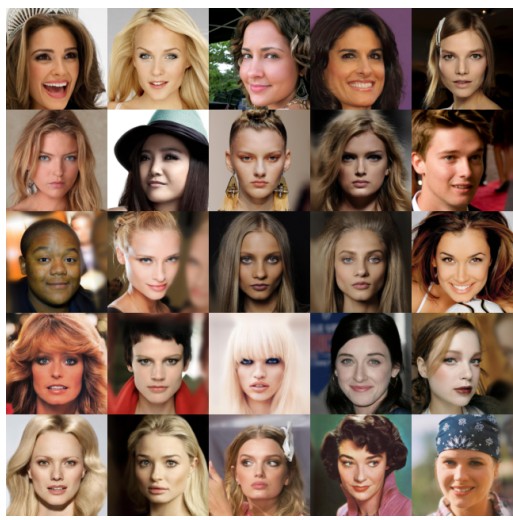 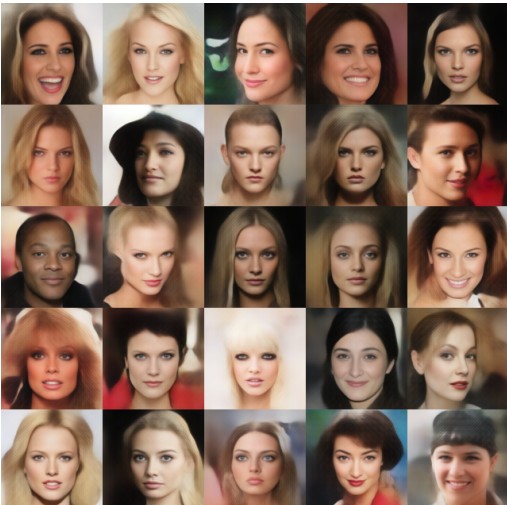

(a) Ground truth test samples        (b) AE reconstructions

Figure 9: AE performance on CelebA-HQ in resolution $128 \times 128$.

## A    NETWORK ARCHITECTURES

### A.1    GENERATIVE MODEL ARCHITECTURE

The generative networks used to address inverse problems described in Section 4 are trained in two-steps: i) we train an auto-encoder (AE) neural network to map the training samples to a lower dimensional latent code, ii) we train a normalizing flow model to learn the distribution of the computed latent codes by using maximum likelihood (ML) loss function.

#### A.1.1    AUTO-ENCODER ARCHITECTURE

The AE network is a succession of an encoder and a decoder neural networks. The encoder maps images to a low dimensional latent code, a $L$-dimensional vector. It is a succession of 6 Conv+ReLu blocks (one convolution layer followed by a ReLu activation unit). Between each Conv+ReLu blocks, there is a down-sampling operation. The decoder network transforms the computed latent code to the image samples. The decoder has the same structure as the encoder network, except that the down-sampling operations are replaced by up-sampling layers. In order to increase the expressivity of the model, we equip our model with local skip connections in both encoder and decoder, as suggested in (He et al., 2016). The latent code dimension is 512 for training data in resolution $128 \times 128$ for both RGB and gray-scale images. The AE model has around 40M and 11M trainable parameters for limited-view CT and CelebA-HQ experiments.

**Training strategy** While we can train the AE model using only the MSE loss (Brehmer & Cranmer, 2020; Kothari et al., 2021), it often suppresses the high-frequency components of the reconstructed image. To alleviate this issue, we combine the perceptual loss proposed in (Johnson et al., 2016) with regular MSE loss which significantly improves the quality of the reconstructions.

**Experimental Result:** We train AE model over 40000 images of the LoDoPaB-CT (Leuschner et al., 2021) and 30000 images of CelebA-HQ (Karras et al., 2017) datasets in resolution $128 \times 128$. We train the model over CT and CelebA-HQ images for 600 and 200 iterations. Figures 10 and 9 display the output image produce by the AE on the test dataset. Notice that very small details are lost by the AE. However, the final reconstruction contains the most important details. We show that this is enough to learn an informative prior and solve ill-posed inverse problems. We also show that it is possible to go beyond the quality of reconstruction given by the trained AE, see Section 4.3.

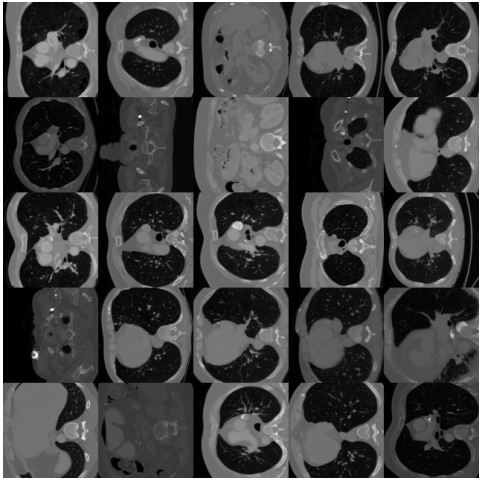

(a) Ground truth test samples

(b) AE reconstructions

Figure 10: AE performance on LoDoPaB-CT in resolution $128 \times 128$.

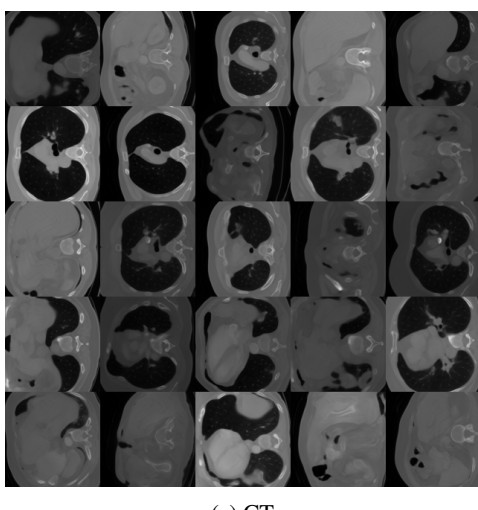
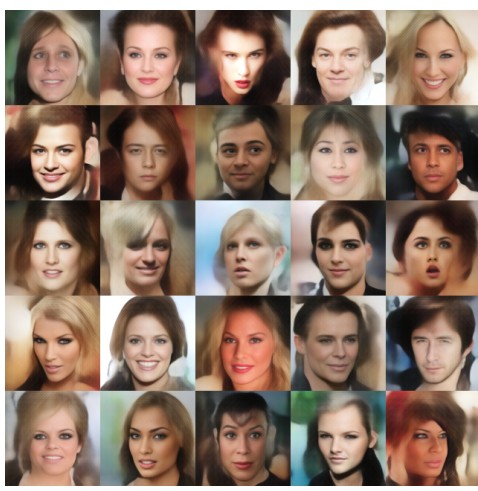

(a) CT

(b) CelebA-HQ

Figure 11: Generative model performance on CT and CelebA-HQ samples in resolution $128 \times 128$.

### A.1.2 NORMALIZING FLOW

We use RealNVP (Dinh et al., 2016) architecture with fully-connected layers for scale and bias subnetworks with two hidden layers of dimension 1024 and 512. The model comprises five masked affine coupling blocks and activation normalization. As mentioned earlier, we train the model using ML loss. The flow model has 10M trainable parameters.

**Experimental Results:** As soon as the flow model is trained, we take samples from the Gaussian distribution and feed to the flow model to generate new latent codes. Decoder then takes the generated latent codes and produce new images. Figure 11 demonstrates the generated samples by our generative model (flows + AE) for CT and CelebA-HQ dataset.

### A.2 FUNKNN ARCHITECTURE

We crop a patch with size $p = 9$ from the low-resolution image by using spatial transformer with *bicubic* interpolation kernel and *reflection* mode for the border pixels. We let two trainable parameters defined the size of the patch with respect to low-resolution image, i.e. the location of

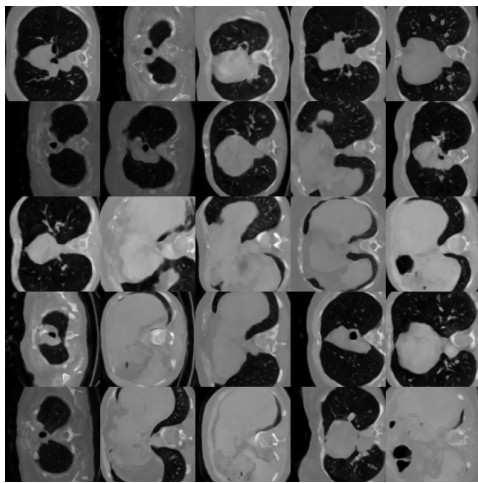

Figure 12: CT images generated by (Dupont et al., 2021)

the 9 pixels around the query pixel are estimated during training. We use eight convolutional layers with filter size 2 and channel dimension 64 and ReLu activation functions. We use two maxpooling layers with size (2,2) inside to reduce the feature size. This downsampling layer if applied between 3 succesive applications of convolution+ReLu layers, so that the size of the input is devided by 4 in total in each direction. We use skip connections between each convolution layers. We then use four fully-connected layers with latent dimension 64 with skip connections between each Linear+Relu block. Finally, following inspiration from residual neural network, we add to the output the intensity of the pixel central to the input patch. We also use mini-batches with 512 pixels and 64 objects for each training iteration. All the models are trained using Adam optimizer (Kingma & Ba, 2014) with learning rate $10^{-4}$.

### A.3  DUPONT ET AL. ARCHITECTURE

We train the continuous generative model proposed in Dupont et al. (Dupont et al., 2021) over the 40000 images of the LoDoPaB-CT dataset. We train their model over 100 epochs using the default parameters given in their public repository. We visually asses the quality of the generated sample in Figure 12.

## B  INVERSE PROBLEMS: EXPERIMENTAL SETTINGS

### B.1  PDE INVERSE PROBLEM

The training strategy of the networks is detailed in Appendix A for two datasets used in Section 5.3. The low-resolution images are composed of $d \times d = 128 \times 128$ pixels and the high-resolutions target images are of size $n \times n = 256 \times 256$. We use $\mathbf{z} = 0$ as the initialization as suggested by (Whang et al., 2021) and optimize for 2500 iterations using Adam optimizer over Problem (3) with $\lambda = 0$ and on Problem (4) with $\lambda = 0$ and $\lambda_2 = 10^{-2}$. Then, we run 1000 iterations of stochastic gradient descent to optimize the weights of the auto-encoder of the generative model as suggested in (Hussein et al., 2020).

## C  LIMITED-VIEW CT

The training strategy of the networks is detailed in Appendix A. The low-resolution images are composed of $d \times d = 128 \times 128$ pixels and the high-resolutions target images are of size $n \times n = 256 \times 256$. We use $\mathbf{z} = 0$ as the initialization. The estimated solution is obtained by running 5000 iterations of stochastic gradient descent using Adam optimization algorithm on Problem (5) with

$\lambda = 0$. The observations given by equation (5) are degraded by additive Gaussian noise such that the SNR of each projection is 30dB.

# D    SUPER-RESOLUTION

Table 2: Performance of different methods over super-resolution in scale-invariant SNR (dB)

|  | $128 \rightarrow 256$ | $128 \rightarrow 512$ | $128 \rightarrow 1024$ | $128 \rightarrow 256$ (OOD) |
|---|---|---|---|---|
| *Bilinear Interpolation* | 27.5 | 24.9 | 22.4 | 22.9 |
| *U-Net (Ronneberger et al., 2015)* | 31.5 | - | - | 22.8 |
| *LIIF-EDSR (Chen et al., 2021)* | 31.6 | 28.0 | 24.0 | **28.0** |
| *LIIF-EDSR (Chen et al., 2021) (140k param.)* | 31.6 | 27.6 | 23.8 | 26.7 |
| *FunkNN (single)* | 31.4 | 26.4 | 22.6 | 27.2 |
| *FunkNN (continuous)* | 30.8 | 27.7 | 23.5 | 26.4 |
| *FunkNN (factor)* | **32.6** | **28.2** | **24.2** | 26.6 |

# E    ACCURACY OF THE DERIVATIVES PROVIDED BY FUNKNN

In order to quantify the quality of FunkNN derivatives, we train our network on images with intensity given by a Gaussian functional:

$$g(x, y) = \exp\left(-\frac{(x - x_0)^2 + (y - y_0)^2}{2\sigma^2}\right),$$

where the position of the center $(x_0, y_0)$ is chosen at random in the field of view and the standard deviation $\sigma$ is chosen uniformly at random in $[0.1, 0.4]$. We use cubic convolutional interpolation kernel Keys (1981) in the spatial transformer of the FunkNN trained using the training procedure described in A.2. In the first row of Figure 13, from left to right, we display the test function $g$, the norm of its gradient and its Laplacian sampled on a discrete grid. in the second row, we display the same quantity estimated by the trained FunkNN as well as the SNR between the estimation and the ground truth.

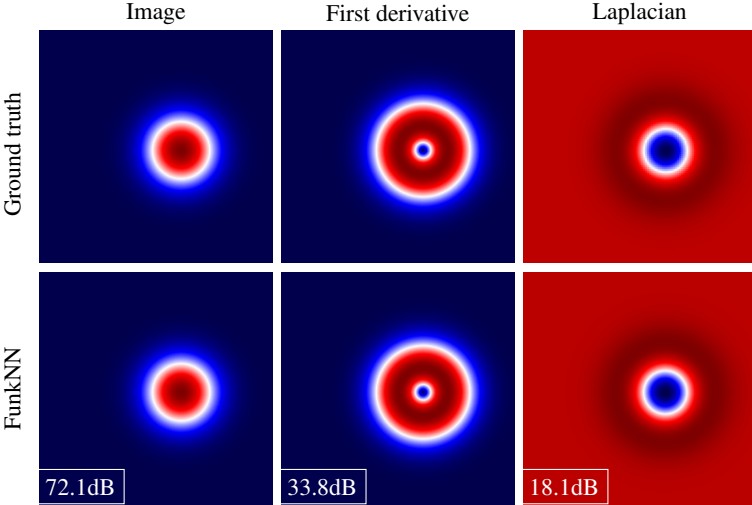

Figure 13: The derivatives produce by FunkNN on images of a Gaussian. The true derivatives can be computed analytically.

