# OpenReview forum: "FunkNN: Neural Interpolation for Functional Generation"
_ICLR.cc/2023/Conference — ICLR 2023 poster_

### Official Review · Reviewer_9a5t · 2022-10-17

**Confidence:** 4
**Correctness:** 3
**Technical Novelty And Significance:** 3
**Empirical Novelty And Significance:** 3
**Recommendation:** 8

**Clarity, Quality, Novelty And Reproducibility:**

**Clarity**

The paper is generally quite clearly written and the figures are nice. A few details are missing in the results section.

**Quality**

The paper is fairly high quality. The method and experiments are both interesting and the paper is quite well written.

**Novelty**

The proposed method is reasonably novel, but is closely related to exisiting methods.

**Reproducibility**

The details and explanations provided in the paper seem to be sufficient for reproducing the results in the paper.

**Details Of Ethics Concerns:**

No ethical concerns.

**Strength And Weaknesses:**

**Strengths**

- The paper tackles an interesting and useful problem.
- The proposed model has various interesting properties. For example, the model can be trained with very low memory since it only requires patches and the continuous nature of the model allows for easy estimation of derivatives.
- The inverse problem applications and experiments are interesting.
- The out of distribution experiments are interesting. It is nice to see that a model trained on CelebAHQ can generalize to LSUN bedrooms.
- The figures are nice and help with the general understanding of the paper.


**Weaknesses**

- In general, the discussion of related work is lacking and the paper is not very well situated in the literature. In particular, the proposed model is very closely related to LIIF which is briefly mentioned as a baseline. However, from reading the FunkNN paper this is not clear. It was not until I had a look at the LIIF paper that I realised how closely related these two ideas are (using local information to upsample images at arbitrary resolutions). The differences need to be discussed in more detail and related works should be properly acknowledged. In addition, generative models of continuous images are briefly mentioned but not discussed in detail (and a reference to [1] is missing). Further, while works like Chen & Zhang 2019 and Dupont et al 2021 are mentioned as performing poorly for grid-based generative tasks, it is worth noting that these models are also much more general and can be applied easily on various data modalities.
- As the proposed method is a super resolution module and not a generative model, I believe it makes more sense to focus on super resolution in the related work section instead of generative models (which currently takes up much more space). In general, the model does not feel very well situated in the super resolution literature.
- There is very little quantitative evaluation of the model. For example, for the inverse problem experiments there is not a single quantitative metric (even though e.g. just measuring reconstruction error should be simple) and only a handful of quanlitative comparisons. Further, there is little discussion of what baselines could be relevant for these experiments.
- The authors discuss generative models throughout the paper, but do not include any experiments or evaluation for this. For example, it would be interesting to see how FunkNN applied to some low res generative model would compare to other generative models of continuous images.
- While the writing is generally clear, there are quite a few typos and some missing information. For example, in Table 1 it is not clear on which dataset the model was evaluated. Is the performance averaged across all examples on some test subset of CelebAHQ?


**Nitpicks**

- It seems like a lot of citations are missing brackets. I would recommend using `\citep{}`.
- The authors often refer to "Implicit Neural Networks". However, it seems like the two dominant terms in the literature now are "Implicit Neural Representations" or "Neural Fields", so I would go with one of these options to be consistent with the literature.

[1] Adversarial Generation of Continuous Images, Skorokhodov et al, CVPR 2021

**Summary Of The Paper:**

This paper introduces FunkNN, a continuous super resolution module for image-like data. Given a low resolution image and an image coordinate x (which may not lie on the low resolution grid), FunkNN returns the predicted RGB value at that coordinate, effectively leading to a continuous image representation, where pixel values can be probed at arbitrary coordinates in R^2. To achieve this, the authors propose to use a spatial transformer which takes as input an image and a coordinate and extracts a (linearly interpolated) patch centered on this coordinate. The patch is then passed through a small CNN which returns the predicted RGB value at the coordinate location. The authors perform various interesting experiments on image super resolution and inverse problems involving image derivatives.

**Summary Of The Review:**

Overall, I think this is a fairly good paper, which may be of interest to the ICLR community. The method and experiments are interesting. However, I do not think the paper is very well situated in the literature and also lacks some experimental evaluation. I therefore believe the paper in its current state is borderline, but I am willing to update my scores if some of the concerns are addressed.

---

> ### Author Response · Authors · 2022-11-19
> **improved discussion of related work + richer quantitative evaluation + all-around clarifications**
>
> Thank you for your detailed review. We made a number of changes to the manuscript to address your critical remarks.
>
> - Discussion of related works: We improved the text to better position our contribution on top of the related works.
> We added a more thorough description of LIIF paper in Section 6.1. We also provided an additional numerical comparison with a lighter version of the LIIF architecture in Section 5.1. which shows that our statements about the relevance of network size are indeed correct.
> A more careful comparison with continuous GANs was also missing. We added a comparison with the continuous GAN proposed by Dupont et al (2021) to solve inverse problems in Section 5.3 and Figure 7. We also reported the generated samples of CT images by Dupont et al. in Figure 13. While the generated samples are promising, their architecture performs poorly combined with gradient-based solvers for inverse problems. We added the corresponding discussionin Section 5.3.
> We have also added a reference to [1] in Section 6 and mentioned that continuous GANs like the model of Dupont et al. are general models and can work on various data modalities.
>
> - Quantitative evaluation: We first compare FunkNN on super-resolution with existing architectures including U-Net and a lighter version of LIIF in more detail. We have added several new measures in Table 1. Regarding inverse problems, we added the SNR of the reconstructions by FunkNN when compared with baselines in figures 7 and 8.
> FunkNN with a generative model: We added a sentence at the end of Section 3.2 and compared our approach to  Dupont et al. for the inverse problem described in Section 4.2. We emphasize that our approach allows us to use any generative model, and in particular normalizing flows that are particularly well suited to solving inverse problems unlike standard GANs like the one in Dupont et al. We showed that even though Dupont et al. can obtain visually good quality samples, their architecture is not well suited to solving inverse problems.
>
> - Unclear Table 1: we wrote a more precise title for Table 1 and expanded the text referring to the table.
> “As the proposed method is a super resolution module and not a generative model, I believe it makes more sense to focus on super resolution in the related work section instead of generative models (which currently takes up much more space). In general, the model does not feel very well situated in the super resolution literature.”: We have substantially rewritten Section 6.1 (super-resolution) and pruned Section 6.2 (generative models) to better place FunkNN in the related literature.
>
> - Thank you also for the ‘nitpicks’ comments which we implemented. We also replaced "Implicit Neural Networks" by "Implicit Neural Representations" in line with the current literature.

---

> > ### Comment · Reviewer_9a5t · 2022-11-20
> > **Thank you for your response**
> >
> > Thank you for your detailed and thorough response. The proposed changes have addressed most of my concerns and have considerably improved the quality of the paper. In particular, the extended discussion and experimental comparisons to LIIF situate the paper much better in the literature and also demonstrate the empirical advantage of using FunkNN over LIIF. Further, I appreciate the added qualitative comparisons for the inverse problems. Based on these changes, I have updated my score from 5 to 8 and would recommend this paper for acceptance.

---

### Official Review · Reviewer_BzrP · 2022-10-20

**Confidence:** 3
**Correctness:** 3
**Technical Novelty And Significance:** 3
**Empirical Novelty And Significance:** Not applicable
**Recommendation:** 8

**Clarity, Quality, Novelty And Reproducibility:**

The paper is clearly written presents the proposed model and test setting with all the necessary details to reproduce its experiments.

The paper take advantages of previous work (spatial transformers) and re-combines it in a novel and well-justified way for the task of super-resolution.

**Strength And Weaknesses:**

**Strengths:**

The paper proposes a simple and elegant model, showing the possibility to build continuous super-resolution methods with *relatively* small models. The efficiency of FunkNN is demonstrated on multiple different tasks, illustrating the various possible applications of the model. The construction of the model is well motivated.

The core idea of the paper is in my opinion the fact that it delegates the "interpolation" part of the problem to a spatial transformer which achieves bilinear interpolation over the whole patch, rather than trying to train a NN to do it. It does seem like a good inductive prior to give to the CNN for the final prediction.

**Weaknesses:**

First of all, I want to say that I am not deeply familiar with the problem of super-resolution, and I refined my understanding of it using the references from the paper. My attention went in particular to the LIIF model (reference Chen et al. 2021 from the paper), which seems to me to have quite a lot of similarities with the proposed FunkNN.

The models are clearly different:
- LIIF passes the whole low-resolution image in a CNN encoder, then extracts a small patch from that encoded representation, and uses a NN to interpolate to the target pixel value from that small patch.
- FunkNN extracts a small patch from the low-resolutions and at the same time interpolates it to align its pixel grid to the target coordinate using a spatial transformer, and then use a CNN to predict the target pixel value from that aligned patch.

However, as far as I can tell, the main properties of FunkNN advanced by the paper (being able to compute spatial gradients of the image and being able to back-propagate gradients to the low-resolution input) are shared by LIIF. This means that for all the inverse problems described in section 4, on which the performance of FunkNN is illustrated, it would be possible to make a performance comparison with LIIF.

Furthermore, FunkNN like LIIF represents the image prediction as a continuous function of pixel values. This means that, like LIIF, FunkNN could theoretically be able to generalize to scale factors different than the ones it was trained on. LIIF for example is trained on x2, x3 and x4 factors, and shown to generalize well up to x30 from that. No such attempt at generalization is made on FunkNN.

The paper points the large number of parameters of LIIF. From what I understand, it comes mostly from the large CNN encoder LIIF uses, that works on 48x48 patches. However, nothing in the structure of LIIF would prevent using a much smaller encoder working on smaller patches (for example 8x8 like FunkNN in the experiments). Doing so could probably bring LIIF to a number of parameters comparable with FunkNN.

The core of my concern is thus this: while conceptually different (interpolation done by a NN vs. done by the spatial transformer), LIIF and FunkNN are two models with very similar sets of capabilities. I thus find it a lack in the paper that the two models are not compared in a more in-depth way: the inverse problems FunkNN is evaluated on should be accessible to LIIF, and the arbitrary scaling of LIIF should be accessible to FunkNN. I find this especially lacking given that, from a conceptual point of view, it seems to me that the good inductive bias of FunkNN has the potential to make it very competitive with LIIR on all those tasks.

**Summary Of The Paper:**

The paper introduces FunkNN, a super-resolution architecture built on a two-stage process to model images as a continuous function.

In order to predict the color value for a given (x, y) location, first a Spatial Transformer with a bilinear kernel is used to extract a PxP patch from the image whose coordinates are centered on the point (x, y), and then a CNN is used to predict the color value at the chosen point. In a sense, one could tell that FunkNN is built by using a CNN to sharpen an initial bilinear upscaling.

The model as a whole allows to both compute spatial gradients in the image, and back-propagate gradient information back to the low-resolution image. This makes it possible to combine FunkNN with any off-the-shelf low-resolution generative model. The paper illustrates this by using such models as a prior over in images in problems of high-resolution image reconstruction from partial data: reconstructing an image from its 20% highest intensity spatial gradients, or in the context of limited-view CT reconstruction.

On the super-resolution task, FunkNN is shown to be competitive to LIIF, a state of the art model, on x2, x4 and x8 tasks, while being a much smaller model in terms of parameter count.

**Summary Of The Review:**

I believe this is overall a decent paper. The proposed model is well-justified, simple and elegant, and experimentally shown to be applicable to several relevant tasks.

However I think that to truly be a good paper it is lacking an in-depth comparison with other models of the literature that can be applied to the same tasks (I focused on LIIR from the references of the paper, but it's entirely possible other models could also be applicable here).

---

> ### Author Response · Authors · 2022-11-19
> **many new comparisons with LIIF**
>
> Thank you for the remarks about comparisons with LIIF. We’ve made a major update to the manuscript and the experiments following your concerns.
>
> - Difference and comparison between LIIF and FunkNN: The key conceptual difference between FunkNN and LIIF is in the way they combine image features and spatial coordinates. LIIF generates image features using an encoder with a large receptive field and then selects features in a small neighbourhood around spatial coordinate. These features along with the coordinate are provided to an MLP to regress the high-resolution image intensity. On the other hand, FunkNN exploits the idea that super-resolving at a coordinate should only require local information. (Provided that we want a stable, robust method.) Therefore, instead of generating features that could depend on global image features and combining them later with the coordinate, FunkNN directly imposes the locality constraint on the input image by (differentiably) cropping a patch around a spatial coordinate and passing it to a small CNN. We have added this discussion to the related work in Section 6.1.
>
>   - For a clearer comparison, we added to Table 1 a “light” version of LIIF architecture so that it has roughly the same number of parameters as FunkNN (140k). This lighter network results in a drop in the quality of the super-resolution results, especially for out-of-distribution evaluation. We also report the evaluation time for both LIIF and FunkNN at the end of Section 5.1.
>
>   - Additionally, the spatial transformer enables us to have a learnable receptive field of the cropper by adding only two learnable parameters (see Section A.2). This new feature significantly improves performance of FunkNN. Finally, the local and global skip-connections in the CNN employed by FunkNN significantly speed up training.
>
> - Reducing the encoder size of LIIF: you are right that LIIF’s parameters can be significantly reduced if the encoder is made smaller with a reduced receptive field. We reduce the number of residual blocks in LIIF’s EDSR encoder from 12 to 2 and obtain parameters comparable to FunkNN. As described before, doing so results in a significant degradation in the super-resolution performance of LIIF.
>
> - Super-resolution factor: In Table 1 and in Figure 4, we demonstrate a super-resolution reconstruction up to a factor 8. An 8-fold increase in resolution with accurately preserved image features in inverse problems is already significant. Even though results of Table 1 suggest that we might perform well on higher super-resolution factors, we choose not to go in this direction since it would require using a different dataset, as the maximum resolution of CelebA-HQ is 1024. This would require re-training the network which we could not do in the given time given all the other new experiments. We rather chose to strengthen the comparisons with other methods since that concern is shared by almost all reviewers.

---

> > ### Comment · Reviewer_BzrP · 2022-11-21
> > **Thank you for your response**
> >
> > Thank you for your detailed response. With the modifications integrated to the paper I believe it is now a solid article that gives a good presentation and analysis of the proposed FunkNN. I am updating my score, and recommend this paper to be accepted.

---

### Official Review · Reviewer_Zsei · 2022-10-21

**Confidence:** 4
**Correctness:** 4
**Technical Novelty And Significance:** 3
**Empirical Novelty And Significance:** 3
**Recommendation:** 6

**Clarity, Quality, Novelty And Reproducibility:**

The paper is clearly written. The proposed approach is moderately novel and well-executed. The authors do not provide code, but I think the paper could be reproduced from the details in the text.

**Strength And Weaknesses:**

Strengths:
1. The proposed method is well-motivated and clearly explained.
2. The method is sensible and has a number of desirable properties.

Weaknesses:
The main weakness is that the experimental evaluation could be more extensive. For example:
1. In "Related Work", the authors discuss GAN-based super-resolution methods, but do not include any such methods as baselines in their experiments.
2. I can imagine that the proposed method could be successfully applied to density estimation. For example, as a cascaded density model similar to what [1] does with diffusion models.

I wonder if more emphasis on directions like these would make the paper more compelling to a broader audience. In contrast, the discussion about OOD generalization feels a bit weak (proposed method not substantially better than baseline), and the image derivative experiments feel a bit niche (I personally do not find these particularly exciting, and imagine that much of the generative modeling community feels similarly).


[1] Cascaded Diffusion Models for High Fidelity Image Generation. Ho et al, https://arxiv.org/abs/2106.15282.

**Summary Of The Paper:**

The authors propose a method for image super-resolution that resembles an implicit function, but has strong inductive biases in the underlying architecture. To predict the super-resolved value for a particular coordinate, they interpolate a small 2D patch centered at that coordinate from the original image (reminiscent of RoI-Align from the segmentation literature), and pass this through a CNN.
They show competitive super-resolution results on natural image datasets, and show that their method produces plausible spatial derivatives that be used in a medical imaging applications.
Compared to other super-resolution methods, theirs seems to be much more efficient in terms of computational complexity and parameter count.

**Summary Of The Review:**

I do think that this paper would be stronger if the authors focused on slightly different directions (see "Weaknesses" section), but overall I feel that it is a solid incremental contribution in super-resolution.

---

> ### Author Response · Authors · 2022-11-19
> **comparisons with continuous GANs, new OOD experiments, PDE-based problems**
>
> Thank you for a positive assessment and constructive suggestions. To better position our contribution in the literature (especially in the literature on generative models), we improved our paper in the following ways:
>
> - We added a comparison with the continuous GAN proposed by Dupont et al. 2021 which performs well for generating continuous images while giving access to the spatial derivatives; see Figure 7 for the derivatives and Figure 13 for the generated samples. In Section 5.3, we show that FunkNN performs significantly better on the inverse problem of image reconstruction from subsampled gradients. Qualitative and quantitative results are reported in Figure 7. This experiment emphasizes the versatility of FunkNN which can be combined with arbitrary discrete generative models.
>
> - Thank you for bringing [1] to our attention; we now refer to it in Section 6. As stated in section 3.2, FunkNN can be combined with arbitrary generative models. In [1] the authors combine diffusion models trained in low-resolution with fixed-size super-resolution networks to generate images in high-resolution. We could replace fixed-size super-resolution by FunkNN to construct a diffusion model for samples at arbitrary resolution. Regarding density estimation, since normalizing flows allow for efficient likelihood computations, we believe FunkNN could be combined with a cheap low-resolution normalizing flow to give likelihoods at any resolution. For reasons of timing, however, the details of this scheme remain an interesting open problem.
>
> - We clarify the position of FunkNN with respect to GANs in Section 3.2. As also mentioned in the response to reviewer kWFU, FunkNN can be trained independently of any generative model.
>
> - Regarding your remark about the weakness of the out-of-distribution experiments: we believe that this experiment conveys a key point. To better articulate it we now added a comparison with the U-Net on super-resolution which clearly shows the strength of FunkNN out of distribution. The experiment shows that the U-Net has comparable results with FunkNN on inlier samples but its performance sharply drops for OOD samples. In general, trained deep-learning models are a priori not expected to perform well on other datasets where their performance usually drops significantly. Due to the simplicity and patch-based processing of FunkNN, it does perform well on out-of-distribution images, while being much more lightweight.
>
> - Inverse problems, especially PDE-based, abound in all scientific domains. They include wave-based problems (medical ultrasound, seismic imaging), diffusion-based problems, electrical impedance tomography, Lorentzian space-time inverse problems, photoacoustic tomography, and many more (see the following papers for more detail). They are traditionally solved by iterative methods and handcrafted priors which limits performance when faced with strong ill-posedness and nonlinearity. FunkNN can be used as a centerpiece of an inversion framework which gives exact derivatives in PDEs while benefiting from a strong generative prior.
>
> **A couple of related references**
>
> - Sitzmann, V., Martel, J., Bergman, A., Lindell, D., & Wetzstein, G. (2020). Implicit neural representations with periodic activation functions. Advances in Neural Information Processing Systems, 33, 7462-7473.
> - Lindell, David B., Julien NP Martel, and Gordon Wetzstein. "Autoint: Automatic integration for fast neural volume rendering." Proceedings of the IEEE/CVF Conference on Computer Vision and Pattern Recognition. 2021.
> - Vlašić, T., Nguyen, H., & Dokmanić, I. (2022). Implicit Neural Representation for Mesh-Free Inverse Obstacle Scattering. arXiv preprint arXiv:2206.02027.
> - Raissi, Maziar, Paris Perdikaris, and George E. Karniadakis. "Physics-informed neural networks: A deep learning framework for solving forward and inverse problems involving nonlinear partial differential equations." Journal of Computational physics 378 (2019): 686-707.

---

### Official Review · Reviewer_kWFU · 2022-11-03

**Confidence:** 4
**Correctness:** 3
**Technical Novelty And Significance:** 3
**Empirical Novelty And Significance:** 3
**Recommendation:** 6

**Clarity, Quality, Novelty And Reproducibility:**

The main approach is simple and easy to follow. However, the method and experiment sections are very hard to follow, with undefined notations and unexplained experimental settings. The contribution of the paper is mainly on the empirical side, and thus more thorough comparisons with state-of-the-art approaches (both quantitatively and qualitatively) on each task are needed. The reproducibility is also unclear.

**Details Of Ethics Concerns:**

Generated super-resolution images could introduce bias if the model is trained on a biased dataset.

**Strength And Weaknesses:**

Strength:
1. The paper is well-written. The proposed approach is simple and easy to follow.
2. The empirical results are good. The model is able to achieve comparable performance while using fewer parameters than state-of-the-art models on several inverse problem tasks.

Weakness:
1. Section 3 is hard to follow. For instance, what does "sample discrete images from a powerful convolutional generator" mean? Do you mean to generate an image using the pretrained generator? What does "discrete image" mean? Does it have a fixed resolution?
2. In section 3.1, it would be better to define the input of ST formally (e.g., coordinate, image) instead of just mentioning its dimension.
3. In section 3.2, does it mean given a pretrained generative model, you apply FunkNN to the output from the generated samples of the model? Why not directly apply it to existing low-resolution images so that the quality of the results will not be bounded by the pretrained generator?
4. The notation is section 4.1 is hard to follow. For instance, a new model IN$_\theta$ is introduced. What is this model doing? It's better to keep the notation consistent. The parameter $j$ is also not formally defined, what are $u$ and $x$?
5. The structure of the paper can be improved. For instance, some results are presented in section 4, but there is another section called Experimental results (section 5).
6.  Besides the super-resolution results (table 1), there is no comparison with state-of-the-art baselines for the other tasks.


**Summary Of The Paper:**

This paper proposes a patch-based framework for solving inverse problems. The proposed approach is able to achieve reasonable performance on several tasks.

**Summary Of The Review:**

This work proposes a simple approach to solving inversion problems. However, the presentation of the paper can be improved, and more empirical results and comparisons with state-of-the-art baselines are needed.

---

> ### Author Response · Authors · 2022-11-19
> **Refactoring, clarifications, and comparisons with new baselines**
>
> Thank you for your refactoring suggestions. We modified the manuscript accordingly and (we believe) considerably improved the presentation.
>
> - “Section 3 is hard to follow.”
>   - We made many changes in Section 3 to streamline presentation. All changes are typeset in blue in the updated manuscript.
>
> - “In section 3.1, it would be better to define the input of ST formally (e.g., coordinate, image) instead of just mentioning its dimension.”:
>   - Agreed; this has been implemented.
>
> - “In section 3.2, does it mean given a pretrained generative model, you apply FunkNN to the output from the generated samples of the model? Why not directly apply it to existing low-resolution images so that the quality of the results will not be bounded by the pretrained generator?”:
>   - It is indeed possible to either apply FunkNN to the output of an arbitrary generative model or to simply train it on a dataset of images, without any generative model. If the goal is to generate continuous samples from a latent space (for example to regularize ill-posed inverse problems), then it makes sense to train FunkNN with a pre-trained generator (or jointly with the generator if a training set is available, although separate training is easier.)
> Per your comment, if the goal is to super-resolve images or their spatial derivatives, then we can directly use low-resolution images during training. We clarify this in the revised manuscript at the end of Section 3.2.
>
> - “The notation in section 4.1 is hard to follow. For instance, a new model INN is introduced. What is this model doing? It's better to keep the notation consistent. The parameter is also not formally defined, what are u and x?”:
>   - We modified the notation in Section 4.1. We replaced ‘INN’ with ‘FunkNN’. The variables u and x are now consistent with Section 3.
>
> - “The structure of the paper can be improved. For instance, some results are presented in section 4, but there is another section called Experimental results (section 5).”:
>   - We now collect all experimental results in Section 5 (the old Section 4 is now subsection 5.3).
>
> - “Besides the super-resolution results (table 1), there is no comparison with state-of-the-art baselines for the other tasks.”
>   - We now provide the following additional comparisons:
> Regarding inverse problems, we added a comparison with the continuous GAN proposed in Dupont et al. (2021) to solve the inverse problem described in Section 4.2. We show that this model behaves poorly with standard gradient-based procedures. Related weaknesses have been previously reported in the literature (cf. Section 5.3). Regarding super-resolution, we added a comparison with the U-Net (Section 5.2) to clearly demonstrate the generalization strength of FunkNN on out-of-distribution samples. We also added an in-depth comparison with the LIIF architecture in Sections 5.1 and 5.2.

---

### Author Response · Authors · 2022-11-19
**Thank you + many updates**

We thank the reviewers for their constructive criticism and actionable comments. In this post we address the concerns shared by all reviewers. Responses to reviewer-specific comments are in the individual replies.

We made the following major improvements:

- Following a suggestion made by several reviewers, we added an extensive experimental comparison between our proposed network, FunkNN, and the current state-of-the-art (SOTA) continuous super-resolution architecture (LIIF).
We previously showed that FunkNN performed comparably to LIIF on continuous super-resolution tasks while having about 10% trainable parameters. To investigate this further, we trained a lighter version of LIIF with a comparable number of parameters. We show that this causes a performance drop for LIIF of up to 1.3 dB. We also show that a forward pass with FunkNN is faster than that of both the original and the lighter versions of LIIF. We argue that the considerably better parameter and compute efficiency of FunkNN over LIIF is a consequence of the different strategies the two networks use to combine image features and spatial coordinates; we argue that the way this is achieved in FunkNN is more natural and straightforward. We discuss this in greater detail in our response to Reviewer BzrP.

- We added a comparison with an existing continuous GAN architecture for solving inverse problems; the comparison shows that continuous GANs are ill-suited for regularization of inverse problems due to limited modeling capacity.. To the contrary, FunkNN can be combined with _any_ existing generative model. We show that combining it with normalizing flows which are well suited for inverse problems yields significantly better performance than continuous GANs.

- In order to emphasize the strong performance of FunkNN on out-of-distribution samples, we compared it with a U-Net which is one of the most successful architectures for image–to–image problems like superresolution but is limited to images of fixed size. We show that FunkNN matches the strong performance of a U-Net on in-distribution images and performs significantly better on out-of-distribution images.

- We made two (significant) updates to further improve the visual quality of FunkNN output. First, we use global and local skip-connections in the convolutional network of FunkNN which speeds up training by a factor of more than 10 and improves the quality of outputs. Second, we let the size of the input patch be estimated during training. This simple modification requires only two additional learnable parameters in the Spatial Transformer while improving performance.

- Finally, we posted the reproducible code in the supplementary material (not submitting it originally was an accidental omission).

---

### Decision · Program_Chairs · 2023-01-20

**Decision:**

Accept: poster

**Justification For Why Not Higher Score:**

Focus is somewhat niche, so harder to see its interest to a broader audience. It's not clear the extent to which the authors fully resolved the reviewers' shared concerns about writing clarity and baselines (unfortunately, little participated in the discussion).

**Justification For Why Not Lower Score:**

The reviewers generally found the paper well-written overall, the idea intuitive and with sensible properties, and with good empirical results.

**Metareview: Summary, Strengths And Weaknesses:**

This paper examines continual signal generation through a signal-processing perspective. They propose FunkNN, a CNN that reconstructs continuous images at arbitrary coordinates and, when combined with a discrete generative model, it can act as a "prior in continuous ill-posed inverse problems." The authors further examine results for high-quality continuous image generation, out-of-distribution performance, and inverse problems.

The reviewers generally found the paper well-written overall, the idea intuitive and with sensible properties, and with good empirical results. There are quite a few gaps in the writing that is pointed out (e.g., related work, lack of clarity in parts of Section 3) and the lack of comparison to state-of-the-art baselines outside the superresolution task. The writing details are easily fixable (and have been according to the authors during rebuttal), and several new baselines have been added during the rebuttal.

All reviewers leaned toward accept and I agree with their consensus.

Note: There was an ethics flag by Reviewer kWFU on potential bias during superresolution. I think this is something independent of the specific paper's contributions and should be handled when applying the method to certain datasets.

**Note From Pc:**

if the above contains the word "oral" or "spotlight" please see: "oral" presentation means -> notable-top-5% and "spotlight" means -> notable-top-25%. As stated in our emails, we are disassociating presentation type from AC recommendations